# The Evolutionary Dance between Innate Host Antiviral Pathways and SARS-CoV-2

**DOI:** 10.3390/pathogens11050538

**Published:** 2022-05-03

**Authors:** Saba R. Aliyari, Natalie Quanquin, Olivier Pernet, Shilei Zhang, Lulan Wang, Genhong Cheng

**Affiliations:** 1Department of Microbiology, Immunology and Molecular Genetics, University of California, Los Angeles, CA 90095, USA; aliyarirs@g.ucla.edu (S.R.A.); hkwp8600@g.ucla.edu (S.Z.); lulanwang@g.ucla.edu (L.W.); 2Department of Pediatrics, Division of Infectious Diseases, Children’s Hospital Los Angeles, Los Angeles, CA 90027, USA; vashkoda@gmail.com; 3EnViro International Laboratories, Los Angeles, CA 90025, USA; opernet@ucla.edu

**Keywords:** innate immunity, interferon, SARS-CoV-2, COVID-19

## Abstract

Compared to what we knew at the start of the SARS-CoV-2 global pandemic, our understanding of the interplay between the interferon signaling pathway and SARS-CoV-2 infection has dramatically increased. Innate antiviral strategies range from the direct inhibition of viral components to reprograming the host’s own metabolic pathways to block viral infection. SARS-CoV-2 has also evolved to exploit diverse tactics to overcome immune barriers and successfully infect host cells. Herein, we review the current knowledge of the innate immune signaling pathways triggered by SARS-CoV-2 with a focus on the type I interferon response, as well as the mechanisms by which SARS-CoV-2 impairs those defenses.

## 1. Introduction

Viruses of the *Coronaviridae* family (CoV) mainly infect the respiratory and gastrointestinal tracts of their hosts, and genetically fall into four distinct genera: *alphacoronaviruses, betacoronaviruses, deltacoronaviruses*, and *gammacoronaviruses* [1]. Mammals are the primary hosts of the first two, whereas the rest mainly infect avian species [2]. Six human CoVs had been identified prior to 2019: HCoV-NL63 and HcoV-229E belonging to the *alphacoronavirus* genus, and HcoV-OC43, HcoV-HKU1, SARS-CoV, and MERS-CoV belonging to the *betacoronavirus* genus (Figure 1) [2]. HcoV-NL63, HcoV-229E, HcoV-OC43, and HcoV-HKU1 infect the upper respiratory tract and cause mild symptoms. However, MERS-CoV, SARS-CoV-1, and SARS-CoV-2 (the latter being a recently discovered seventh human CoV) cause lower respiratory tract infections with estimated fatality rates of 30%, 10%, and 1.25%, respectively.

Severe acute respiratory syndrome coronavirus 2 (SARS-CoV-2) is the etiological agent causing the current global pandemic of coronavirus disease 2019 (COVID-19). It is an enveloped, positive-sense RNA virus with a 29.9 kb genome that is a full length mRNA copy with a 5′ cap and a 3′ polyadenylated tail. The genomic RNA is protected by a helicoidal capsid made of nucleocapsid (N) protein, and is further surrounded by an envelope stabilized by the E protein and a membrane (M) protein anchoring the surface spike (S) glycoprotein. SARS-CoV-2 binds the ACE2 receptor and fuses with the cell membrane. The plus-sense viral RNA genome is released and directly translated into polyproteins 1a and 1ab. These are cleaved to form multiple nonstructural proteins (nsps) and RNA-dependent RNA polymerase (RdRp). RdRp produces the genome, anti-genome, and sub-genomic mRNAs that encode the remaining viral proteins, including N, M, E, and S. The nucleocapsid assembles with the genome copies while the other proteins are modified and escape into a ER–Golgi-intermediate complex, which then come together to form a mature virion that is released by exocytosis (Figure 1). Due to several factors, including a high infectivity rate, a mutable surface antigen resulting in multiple variants, inflammatory properties leading to autoimmune and prolonged symptoms after infection, a wide spectrum of disease presentations with a large number of asymptomatic carriers, as well as limited and short-lasting immune memory responses, SARS-CoV-2 has become one of the most challenging public health threats in decades.

The innate immune system is the host’s first line of defense and provides a critical and rapid response against invading pathogens. It recruits cells carrying germline-encoded pattern recognition receptors (PRRs) that detect pathogen-associated molecular patterns (PAMPs). PAMPs are highly conserved motifs of microbial origin that activate host antiviral pathways. The interferon (IFN) response is considered the primary antiviral innate immune signaling pathway in the host, and leads to the production of interferons that are categorized into three types. Type I interferon (IFN-I) and type III (IFN-III) in turn activate interferon-stimulated genes (ISGs), the products of which place the host on alert in an antiviral state. In addition, individual IFN-I members (α and β) and type II IFN (IFN-II) (γ) can directly downregulate certain host genes to inhibit viral infections [3]. Among the host’s PRRs are those that detect virus-derived nucleic acids generated during viral replication in the cytoplasm [4,5,6,7,8]. These include viral RNA sensors such as the RIG-I-like family members RIG-I and MDA-5, or viral DNA sensors such as cGAS, DDX41, and IFI16 [9,10,11]. Microbial nucleic acids detected in the endosomal compartments or on the cell surface activate the host IFN-I response through membrane-bound Toll-like receptors (TLRs) -3, -7, -8, and -9 [9,12,13]. The IFN-I signaling pathway is also induced by TLR4 upon the detection of lipopolysaccharide (LPS) derived from the cell walls of Gram-negative bacteria [14]. IFN-I plays a critical role in the activation of the antiviral state [15,16]. It acts in autocrine and paracrine fashions by binding to the IFNα/β receptor (IFNAR) on neighboring cells and inducing the Janus kinase-signal transducer and activator of transcription (JAK-STAT) signaling pathway, which subsequently upregulates the expression of a few hundred ISGs [17,18,19,20]. These ISGs operate either directly by targeting viral components or indirectly by interacting with cellular pathways to inhibit viruses at different stages of their life cycle [21]. In addition, individual IFN-I members (α and β) and IFN-II (γ) [22] can downregulate certain host genes to inhibit viral infections [3]. Patients with mild COVID-19 symptoms were found to have elevated ISG expression in multiple white blood cell lines that were absent in those with severe disease. Some patients with severe COVID-19 were even found to have pre-existing autoantibodies to IFN-I, which lowered their ability to combat the virus [23]. Pediatric patients with COVID-19, who have better outcomes than adults, were also found to have higher levels of some innate molecules and increased innate immune responses compared to older patients [24]. Mice lacking the IFNAR receptor have increased susceptibility to SARS-CoV-2 [25]. Supplemental IFN-I has been tested as a treatment for SARS-CoV-1, MERS-CoV, and SARS-CoV-2. While IFN-α and -β were shown to reduce mortality if given early in severe COVID cases, the late administration of exogenous IFN-α actually increased the likelihood of death [26]. This agrees with the commonly accepted belief that IFN-I can play both a protective and pathogenic role in different infections [27]. IFN-I members are expressed in a positive feedback loop, but the downstream JAK/STAT pathway contains a negative feedback mechanism. If these are not in balance, the pathogenic effects of the virus or host can be devastating [28]. SARS-CoV-2, similar to many other viruses, has evolved mechanisms to evade or counteract the host innate immune response.

## 2. Unique Properties of SARS-CoV-2

SARS-CoV-2 falls into the same *betacoronavirus* genus as the highly pathogenic viruses SARS-CoV-1 and MERS-CoV, but this group also includes the milder HCoV-OC43 and HCoV-HKU1 (Figure 2) [29]. Despite their common lineage, these viruses each carry multiple and sometimes significant changes in their genome organization and protein structures [30,31]. This could be related to adaptations to different intermediate animal hosts that served as a bridge between what were originally bat viruses and have since also become human pathogens [32]. For instance, the spike surface glycoprotein (S), which has both a host cell receptor binding domain (S1) and a membrane fusion domain (S2), has a unique furin cleavage site in SARS-CoV-2 between S1 and S2 not present in other CoVs that facilitates viral entry and infectivity [32,33]. Despite SARS-CoV-1 and SARS-CoV-2 both binding to angiotensin converting enzyme 2 (ACE2) and CD209 L receptors in the host, the latter virus was also found to bind to NRP1 [33], PIKfyve, and BSG [34]. This might explain why COVID-19 causes more than just respiratory symptoms [30,35], being noted also for gastrointestinal and even neurological features such as loss of taste (ageusia) and smell (anosmia). These two closely related viruses were even found to interfere with host immune responses through different mechanisms [30,36,37,38,39,40,41,42]. It is therefore not surprising that many therapeutics that showed promise in SARS-CoV-1 infection, such as the HIV protease inhibitors lopinavir and ritonavir [43] or combined therapy with IFN-α and the broad antiviral ribavirin [44], had no impact on COVID-19 [45,46], with further drug combinations required to see an effect [47].

While the original SARS pandemic infected over 8000 people in 29 countries between 2002 and 2004, human cases disappeared suddenly. This was thought to be due to a combination of factors, including improved public awareness and sanitary practices and quarantining of infected humans and animals, to limit the spread [30]. SARS-CoV-1 cases were largely symptomatic with fever and respiratory problems at the early stages of infection when the virus was contagious. This is in sharp contrast to SARS-CoV-2, where about 40% of healthy hosts have no or only mild symptoms [48] that present anywhere from 2–14 days after the initial infection. The high viral load in the upper respiratory tract of even asymptomatic adults [49] essentially marks anyone as a potential source of exposure [50], although young children are considered far less likely to be spreaders [51,52]. Children also have a higher tendency to be asymptomatic [52,53], with possible explanations ranging from their frequent exposure to other coronaviruses or other vaccines granting cross-reactive immunity [48,53], to a more limited distribution of ACE2 and other SARS-CoV-2 receptors, and a tendency to have fewer co-morbidities [35] associated with the progression of infection to severe pneumonia, respiratory distress requiring supplemental oxygen support, coagulopathies, and organ failure. Those with obesity, diabetes, hypertension or chronic lung, heart, or kidney issues are at higher risk and represent the majority of COVID-19 cases requiring hospital admission [54].

The symptoms of COVID-19 initially arise from direct viral invasion of host tissues, but in some, the disease progresses largely due to the host’s own immune response [38,55]. Cough, rhinorrhea, and pharyngitis are related to increased nasal secretions, airway inflammation, and immune cell infiltration into local tissues [56,57]. Gastrointestinal symptoms may be related to immune cell infiltration and a disruption in bowel wall integrity where ACE2 is expressed [58]. The ageusia and anosmia seen with COVID-19 are likely to result from damage to the cells that support olfactory neurons, which also carry ACE2 [59]. However, fever, fatigue, chills and myalgias, which are common responses to many viral infections and even vaccinations, are the result of pro-inflammatory cytokines, including those of the host interferon response [56]. These are an outward sign of the body being in an activated antiviral state. This provides some protection in early stages of infection; however, when over-amplified or prolonged, it leads to a cytokine storm effect [36,60] that triggers lung damage and edema (acute respiratory distress syndrome) and disseminated intravascular coagulation (DIC) that can lead to stroke, pulmonary embolism, and organ ischemia. These are the root causes of the morbidity and mortality seen in acute COVID-19 infection [61].

Although children without co-morbidities are usually spared the more severe outcomes of COVID-19 [52], a history of exposure to the S protein, whether from SARS-CoV-2 infection or vaccination [62], puts them as risk of a separate clinical entity known as Multi-Inflammatory Syndrome (MIS). Although some cases have been reported in adults, it is far more common in children (MIS-C) [63], yet still considered relatively rare. This is an autoimmune reaction believed to be triggered by molecular mimicry between the S protein and host antigens on vascular or cardiac tissue [64]. Immunoglobulin G (IgG) to the S protein forms between 2 to 6 weeks after the initial exposure; in some children, this will then proceed to cause a highly inflammatory syndrome that includes sequelae of vasculitis (rashes, conjunctivitis, oral mucosal changes), lymphadenitis, gastrointestinal symptoms, coagulopathy, and myocarditis [63,65]. Many of these symptoms overlap with those of acute SARS-CoV-2 infection, although the underlying mechanisms differ. In both severe COVID-19 and MIS-C, suppression of the overactive immune response with corticosteroids has become standard care [63], although MIS-C often also requires the neutralization of autoantibodies with intravenous immunoglobulin or specific antagonists to IL-1 (anakinra) or IL-6 (tocilizumab—although its use is no longer recommended) [64,65,66].

## 3. Innate Immune Detection of SARS-CoV-2

The innate immune system is triggered by PRRs, which have evolved to quickly recognize viral PAMPs in order to limit further microbial invasion. There are distinct but overlapping PRRs for detecting SARS-CoV-2-derived PAMPs, including viral particles and their derivative molecules and damage-associated proteins (DAMPs), which are released from damaged tissue upon viral attack. The primary PRRs involved in sensing invading pathogens are Toll-like receptors (TLRs), retinoic acid-inducible gene 1 (RIG-1)-like receptors (RLRs), nucleotide-binding oligomerization domain (NOD)-like receptors (NLRs), and C-type lectin receptors (CLRs). The detection of viral PAMPs by PRRS results in the activation of downstream signaling pathways, which subsequently promote the expression of a wide variety of cytokines and chemokines, including interferons. IFN-I and IFN-III regulate the expression of multiple innate immune response genes that place the host in a state of antiviral preparedness. Below, we briefly review the roles of TLRs and RLRs in response to SARS-CoV-2 infection; these are summarized in Figure 3.

### 3.1. Toll-Like Receptors

TLRs are membrane-bound receptors that play a crucial role in the recognition of viral PAMPs. Twelve TLRs have been identified, of which TLR2 and TLR4 target viral proteins and TLR3, TLR7, TLR8, and TLR9 target viral nucleic acids [14]. TLR2 recognition of the SARS-CoV-2 E protein promotes the expression of pro-inflammatory genes in an entry-independent manner [67]. In patients with severe COVID-19 disease, TLR2 was found to be significantly induced compared to patients with only moderate symptoms [68]. The administration of a TLR2-inhibitor was also partially protective against SARS-CoV-2-induced symptoms in mice [67].

TLR3 recognizes viral dsRNA intermediates generated transiently upon viral replication. This step plays an important role in blocking infection by a wide variety of RNA viruses, including SARS-CoV-2. Mutations in genes involved in the TLR3 signaling pathway are seen in 3.5% of patients with severe COVID-19 clinical manifestations [69]. Mice lacking the TRIF adaptor of TLR3 experienced greater mortality from SARS-CoV-1 infection than wild-type controls [70]. Despite this, there was no significant difference in mortality between wild-type and TLR3-knockout mice infected with SARS-CoV-1.

TLR4 is bound to the cell membrane and has been extensively studied for its recognition of numerous PAMPs, including LPS, viral envelope proteins, and DAMPs [71]. In an attempt to identify the factors responsible for the cytokine storm seen in severe cases of COVID-19, genes involved in the TLR4 pathway were found to be highly upregulated. In addition, the expression of recombinant SARS-CoV-2 S2 and N proteins in human peripheral blood mononuclear cells elicited an inflammatory response similar to that seen in bacterial sepsis [72]. TLR4′s interaction with S2 could induce inflammatory responses in a receptor-independent manner, and is notable for triggering a significant increase in IL-1β [73]. In addition, the S, ORF3-a, and N proteins of SARS-CoV-2 elicit inflammasome activation and interact with NLRP3 to trigger further inflammation [74]. Elevated levels of IL6 and TNFα are also reported in severe COVID-19 cases [68]. The inhibition of TLR4 or IL-1β has therefore been proposed as a mechanism to alleviate COVID-19 symptoms.

Endosomal receptors TLR7 and TLR8 mediate the recognition of viral single-stranded RNAs. The whole exome sequencing of four previously healthy men with severe COVID-19 also revealed the presence of putative loss-of-function variants in X-chromosomal TLR7. In addition, missense TLR7 variants in young men with severe COVID-19 were also reported by another group [75]. These studies suggest an important antiviral role for TLR7 in SARS-CoV-2 infection [76].

### 3.2. RIG-1-Like Receptors

RLRs are a family of cytosolic PRRs that detect foreign RNAs; including laboratory of genetics and physiology protein 2 (LGP2), retinoic acid-inducible gene I (RIG-I), and melanoma differentiation-associated gene 5 (MDA5). RIG-I and MDA5 preferentially recognize distinct virus-derived RNA substrates to trigger IFN-I and IFN-III signaling. Where RIG-I is involved in the recognition of short viral RNA molecules, MDA5 favorably binds long dsRNAs (>1 kb) generated during viral replication. Upon binding to the substrate RNA, MDA-5/RIG-I is activated and signals through adaptor mitochondrial antiviral-signaling proteins (MAVS) on the mitochondrial membrane. The activation of MAVS triggers a series of protein modification events, including ubiquitination by TRAF family members such as TRAF3 and phosphorylation by TBK and IKKε, which allows the homodimerization and nuclear translocation of IRF3/IRF7 [14]. Activated IRF3/IRF7 promotes the transcription of IFN-I and IFN-III to upregulate the expression of ISGs. Several studies have shown evidence of varying roles for RLR in the host antiviral response to SARS-CoV-2, possibly due to differences in their experimental settings [77]. Yamada et al. found that at early stages of infection, the helicase domain of RIG-I binds the 3′UTR of the SARS-CoV-2 genome and interferes with the viral RdRp to prevent the generation of negative-sense genomic RNA from plus-sense SARS-CoV-2. In this scenario, SARS-CoV-2 replication is restricted in a MAVS-independent manner and is not associated with the induction of IFN-I and IFN-III expression [77]. Rebendenne et al. reported the recognition of SARS-CoV-2 RNA by MDA-5, but not by RIG-I, in Calu-3 cells, which resulted in IFN-I and IFN-III expression. However, the MDA-5-mediated enhancement of endogenous IFN-I and IFN-III did not have any inhibitory effect on viral replication. The SARS-CoV-2 viral load in Calu-3 cells was not affected by knocking down MDA5, MAVS, IRF3, or STAT-1; however, pretreatment of primary human airway epithelial (HAE) cells and Calu-3 cells with IFN-I at a dose equivalent to that induced in natural infection, resulted in a 1.5–2-fold decrease in the viral load [78]. The results from this study may partially explain the importance of timing in the ability of the IFN-I immune responses to control SARS-CoV-2 infection. In addition, single-cell transcriptional analysis of upper respiratory samples taken from SARS-CoV-2-infected patients of different age ranges showed that children have the highest basal levels of RIG-I and MDA-5. This may correlate with more potent innate immune signaling in children compared to infected adults [79].

## 4. The Role of SARS-CoV-2 Proteins in Disrupting Innate Immune Responses

The PRRs of the innate immune system have evolved to recognize viral PAMPs and restrict their invasion of host cells. In response, SARS-CoV-2 has adapted secondary functions for many of its proteins (Table 1) in order to subvert the host immune response (Figure 4). These include evading PRR recognition, blocking PRR-mediated signaling cascades, the modulation of ubiquitination and deubiquitination, viral protease-mediated cleavage, and shutting down host translation [80].

The S protein of SARS-CoV-2 is the primary virulence factor of the virus. In addition to directly mediating cell invasion, the S protein also interferes with host cell translation to limit the antiviral response [81]. While the S2 domain is anchored to the viral membrane and is fairly conserved, the S1 domain projects outward and is exposed to host antibodies, conferring selective pressure to mutate. Multiple variant strains have emerged since the original genetic sequence (Wuhan-Hu-1) was released by Chinese scientists in early 2020—the template that was used for the three currently FDA-approved COVID-19 vaccines [82,83]. The appearance of new mutations conferring advantages in establishing infection and evading the immune system has continued even during the pandemic, leading to novel variants [84]. Five of these new strains have been considered “variants of concern”. Noteworthy S1 mutations include D614G, which results in higher viral burdens without affecting disease severity [85]; E484K, which increases antibody escape [86]; and multiple mutations that increase binding specificity to ACE2, including N439K [87], Y453F, S477N, and N501Y [88]. Other insertions or deletions in the S protein include Δ69/70 in the α-variant and 215EPE in the ο variant.

The outer membrane (M) protein of SARS-CoV-2 binds to the viral nucleocapsid (N) protein during viral assembly, and its expression was found to be higher in SARS-CoV-1 than in SARS-CoV-2 [89]. The M protein of SARS-coronavirus (SARS-CoV) was shown to have anti-IFN activity [86,90,91] by targeting IKKε/TBK1, preventing TRAF-3 activation and downstream signaling, including IRF3 activation. It also interferes with MAVS aggregation and RIG-I signaling [92]. Sui et al. also shown that the SARS-CoV-2 M protein inhibited IFN-I expression and TBK1-mediated IRF3 activation. It inhibits the expression of ISGs induced through activation of the RLR and NF-κB pathways via the ubiquitin-dependent degradation of TBK1 [91]. Zhang et al. found that SARS-CoV-2 M protein interacts with multiple host factors including karyopherinα1-6 (KPNA1-6), which is required for the translocation of phosphorylated IRF3, IRF7, and STAT1 from the cytoplasm to the nucleus. M protein interference with IRF3 binding to KPNA6 also inhibits the nuclear translocation of IRF3 [93]. The SARS-CoV-2 N protein was shown to interact with granule assembly factor 1 (G3BP1) and granule assembly factor 2 (G3BP2) to inhibit translation, limiting the expression of ISGs [94]. It is also an inhibitor of RIG-I ubiquitination [95].

The 16 nonstructural proteins (nsps) that are encoded by ORF1ab play important roles in regulating viral replication, including the formation of the viral replicase-transcriptase complex (RTC). Other viral proteins were discovered to interfere with IFN signaling by disrupting the TANK/TBK1/IKK, IFR3/IRF7, and STAT1/STAT2 complexes. SARS-CoV-2 ORF1b utilizes −1 programmed ribosomal frameshifting (PRF), which permits the translation of a number of viral proteins encoded in alternative open reading frames (ORFs) from a single template. ORF1 of the SARS-CoV-2 RNA genome contains a pseudoknot with a stop codon and allows for −1 PRF. The efficiency of PRF is estimated to be between 25–75% [96] in SARS-coronaviruses depending on the cell type temperature, and passage number of cells. Kelly et al. have shown that the efficiency of SARS-CoV-2 PRF in rabbit reticulocytes is 36% ± 3%. Therefore, in about 36% of translation events, there is a frame shift that bypasses the stop codon in ORF1 and creates the longer 1ab polyprotein, which additionally encodes nsp 12–16 [97]. Nsp3 in SARS-CoV-1, MERS-CoV, and SARS-CoV-2 contains a single papain-like proteinase (PLpro) domain, whereas other coronaviruses (NL63, HKU, OC43) have two PLpros (PLP1 and PLP2) [98]. Nsp3 is responsible for the cleavage and release of nsp1, nsp2, and nsp3 from the viral polyprotein. Nsp5, a chemotrypsin-like (3C-like) proteinase also known as 3CL^pro^ or M^pro^, cleaves the remaining nsps. Although the primary function of nsp3 and nsp5 is to process the viral polyprotein in a coordinated manner, they have the additional roles of disrupting innate immune responses through cleavage, de-ubiquitination, and de-ISGylation [99,100]. For example, nsp5 targets the host immune response by interfering with RIG-I ubiquitination and also disrupts the formation of antiviral stress granules (avSG) [101,102]. It also cleaves RFN20, which is an antiviral ISG, as well as NLRP12 and TAB1, which are two important regulators of inflammation, contributing to abnormal cytokine expression and general immune dysregulation [80,103].

Nsp1 binds to the mRNA channel on ribosomes to suppress the expression of host proteins [104,105,106] and inhibits the nuclear export of cellular mRNA [107], redirecting host machinery to synthesize viral mRNA [108]. Nsp1 also interferes with the IFN-I pathway [109]. Recent studies have shown that nsp1 expression in A549 cells decreases the expression levels of TLR2, TLR4, and TLR9 [110]. Nsp6 prevents TBK1 targeting and phosphorylating IRF3 [109], which blocks downstream signaling in the IFN-I pathway. Nsp7 is a non-enzymatic subunit of RNA-dependent RNA polymerase (RdRp) that has been shown to interfere with IFN-I signaling [109]. Nsp8 is another RdRp subunit with primase activity. It has an indirect role in inhibiting the innate immune response as it interferes with protein trafficking by targeting 7SL RNA, SRP19, SRP54, and SRP72 [94]. Nsp8 interferes with protein trafficking to the plasma membrane by interacting with 7SL RNA [104] Nsp9 has been shown to bind MIB1, an ubiquitin ligase important for innate immune signaling [94]. Similar to nsp8, it interferes with 7SL and protein trafficking [104]. In addition, nsp9 also interferes with nuclear transport by reducing the amount of nucleoporin 62 (NUP62) on the nuclear membrane [111]. Nsp10 is a modulator of the transcription/replication complex in coronaviruses [112]. Nsp14 is an exonuclease that interacts with nsp10, which is critical for proofreading, and also has N7-methyltransferase activity. Nsp14 requires nsp10 to boost its enzymatic activity, and the deletion of nsp10 was found to prevent SARS-CoV-1 replication [113]. Interestingly, nsp14 and nsp10 have also been shown to interfere with the host immune response through exonuclease activity affecting the translation and limiting the expression of ISGs, including viperin, TRIM21, ISG15, RIG-I, MDA5, and STING [113]. Nsp12 is the main component of RdRp, but has also shown inhibitory effects on the IFN-I pathway [114]. Nsp13 is an RNA helicase that targets TBK1 and TBKBP1, components of the innate immune system [94,109,114,115,116]. Nsp13 also interferes with the NF-κB pathway by interacting with TLE1, TLE3, and TLE5 [94,115]. Nsp15 is an endonuclease that targets the IFN-I pathway by interacting with RNF41/NRDP1 [94] and decreasing the level of TLR2 [110]. Nsp16 is an O-methyl transferase that works with nsp14 to cap viral mRNA to allow efficient translation while also avoiding degradation and recognition by the host innate immune response. It also interferes with the splicing machinery [104] by targeting U1 and U2 mRNA.

The accessory proteins encoded by the remaining ORFs in SARS-CoV-2 have also been shown to play a role in suppressing the host’s antiviral response. ORF3a is a viroporin that binds TRIM59, an ubiquitin ligase involved in innate immune signaling [94]. In SARS-CoV-1 and -2, ORF3b also prevents IRF3 translocation to the nucleus [92]. ORF6 localizes on nuclear membranes and annulate lamellae, co-localizing with nuclear pore complexes (NPC) by interacting with Nup98/Rae-1. It also interacts with KPNB1-KPNA [114,117,118], the karyopherin complex that carries STAT1/2 and other molecules through NPCs upon their activation by IFNAR. This double interaction prevents STAT1/2 from entering the nucleus and activating ISGs, similar to the mechanism observed with vesicular stomatitis virus (VSV) [94,117]. In this way, ORF6 is involved in IFN-I pathway inhibition [109,119]. ORF7a inhibits protein synthesis and activates mitogen p38 [120]. ORF7a and 7b also both interfere with STAT1 and 2 [121,122]. ORF8 has two genotypes (L and S) and is an IFN-I antagonist in both SARS-CoV-1 and SARS-CoV-2. Both genotypes induce ER stress, interfere with IRF3 nuclear translocation, and limit the production of IFNβ [123]. ORF9b inhibits the K63 poly-ubiquitination of NEMO, blocks the interaction between TOM70 and HSP-90, and also interferes with RIG-I, MDA5, MAVS, TBK1, STING, and TRIF [94,124]. ORF9c targets the NF-κB pathway by interacting with NLR family member X1 (NLRX1), F2R like trypsin receptor 1 (F2RL1), and Nedd4 family interacting protein 2 (NDFIP2) [94].

**Table 1 pathogens-11-00538-t001:** Mechanisms of viral interference with host innate immune responses.

Protein	Target	Activity	Virus	Ref
nsp1	NXF1-NXT1	prevents cellular mRNA nuclear export	SARS-CoV-2	[73,125]
	small ribosomal subunit	obstructs RNA tunnel on ribosome	SARS-CoV-2	[87,104,105,106,108,109,110]
	IRF3	prevents dimerization	SARS-CoV-2	[92,109]
	TLR2,4, and 9	alters protein expression	SARS-CoV-2	[110]
nsp2	IRF3	prevents phosphorylation	SARS-CoV-1	[126]
nsp3/PLpro	IRF3	prevents dimerization	SARS-CoV-1	[92]
	IRF3	prevents nuclear translocation	SARS-CoV-1; MERS-CoV	[92]
	IRF3	IRF3 cleavage	SARS-CoV-2	[103]
	ISG15 and ISGylation	reverses ISGylation	SARS-CoV-2	[127,128]
	MDA5	de-ISGylation	SARS-CoV-2	[129]
nsp3/PLP2-TM	IRF3	prevents nuclear translocation	hCoV NL63	[92]
nsp5	NLRP12	aberrant cytokine expression/upregulation	SARS-CoV-2	[103]
	TAB1	aberrant cytokine expression/upregulation	SARS-CoV-2	[103]
	RIG-I-MAVS	targets ubiquitination of RIG-I	SARS-CoV-2	[101,102]
	avSG	prevents formation of stress granules	SARS-CoV-2	[102]
nsp6	TBK1/IRF3	prevents phosporylation of IRF3	SARS-CoV-2	[109]
nsp7	IFN pathway	inhibits the IFN pathway	SARS-CoV-2	[109]
nsp8	7SL, SRP19, SRP54 and SRP72	interferes with protein trafficking	SARS-CoV-2	[94,104]
nsp9	MIB1	blocks ubiquitination/innate immune signaling	SARS-CoV-2	[94]
	7SL	interferes with protein trafficking	SARS-CoV-2	[104]
	NUP62	interferes with nuclear trafficking	SARS-CoV-2	[111]
nsp10	Viperin, TRIM21, ISG15, RIG-I, MDA5, STING	inhibits translation	SARS-CoV-2	[113]
nsp12	IFNB	inhibits the IFN pathway	SARS-CoV-2	[114]
nsp13	TBK1 and TBKBP1, TLE1, TLE3, and TLE5	inhibits the IFN pathway inhibits the NF-κB pathway	SARS-CoV-2 SARS-CoV-2	[94,109,114,115,116]
nsp14	Viperin, TRIM21, ISG15, RIG-I, MDA5, STING	translation shutdown	SARS-CoV-2	[113]
	IRF3	prevents nuclear translocation	SARS-CoV-2	[130]
	IFIT1	viral RNA capping		[130]
nsp15	RNF41	inhibits the IFN pathway	SARS-CoV-2	[94]
	TLR2		SARS-CoV-2	[110]
nsp16	U1/U2 snRNA	interferes with host mRNA splicing	SARS-CoV-2	[104]
S	ELF3f	interferes with host translation	SARS-CoV-1	[131]
M	TRAF3/IKKε/TBK1	prevents the formation of TRAF3·TANK·TBK1/IKKε complex	SARS-CoV-1	[89,92]
	TRAF3/IKKε/TBK1	prevents the formation of TRAF3·TANK·TBK1/IKKε complex	SARS-CoV-2	[89,92,114]
N	IRF3	prevents dimerization	SARS-CoV-1	[92]
	IFN-I	general inhibition	SARS-CoV-2	[119]
	G3BP1 and G3BP2	inhibits host translation	SARS-CoV-2	[94]
ORF3a	TRIM59/STAT1	inhibits STAT1 phosphorylation	SARS-CoV-2	[94]
ORF3b	IRF3	prevents nuclear translocation	SARS-CoV-1	[92]
ORF4a	IRF3	prevents nuclear translocation	MERS-CoV	[92]
ORF4b	TRAF3/IKKε/TBK1		MERS-CoV	[92]
ORF5	IRF3	prevents nuclear translocation	MERS-CoV	[92]
ORF6	IRF3	prevents nuclear translocation	SARS-CoV-1	[92]
	IRF3	prevents nuclear translocation	SARS-CoV-2	[92]
	IRF3	prevents activation/dimerization	SARS-CoV-2	[92,109]
	Nup98/Rae1—STAT	blocks the nuclear pore and prevents STAT1/2 nuclear translocation	SARS-CoV-2	[94,114,117,118]
	IFN-I	inhibits the IFN-I pathway	SARS-CoV-2	[119]
ORF7a	p38	activation of p38 (mitogen)	SARS-CoV-1	[120]
ORF8	IRF3	prevents dimerization	SARS-CoV-1	[92]
	IRF3	prevents nuclear translocation	SARS-CoV-2	[123]
	IFNβ via ER stress		SARS-CoV-2	[123]
	IFN-I	general IFN-pathway inhibition	SARS-CoV-2	[119]
ORF9b	NEMO, TOM70, RIG-I, MDA5, MAVS, TBK1	inhibits the IFN pathway	SARS-CoV-2	[94,124]
ORF9c	NLRX1, F2RL1, and NDFIP2	inhibits the NF-κB pathway	SARS-CoV-2	[94]

## 5. Interferon-Stimulated Genes

The detection of pathogenic components by TLRs and RLRs results in the induction of IFN-I and IFN-III, which subsequently regulate the expression of a few hundred ISGs [19]. However, only particular subsets of ISGs appear to be required against specific viruses. An ISG may naturally have diverse or even opposite functions in different cells or against different pathogens. In addition to ISGs, interferons may regulate the antiviral state by reprogramming host metabolic pathways to combat viral infection. However, the uncontrolled overexpression of cytokines promotes a hyperinflammatory state that can result in massive tissue damage. This includes sepsis-like presentations and the acute-respiratory distress syndrome seen in severe COVID-19 [132]. Therefore, negative feedback of IFN pathways and a return to homeostasis is equally important as controlling infection. Below, we briefly review the roles of several notable ISGs in SARS-CoV-2 infection.

### 5.1. ACE2

The coronaviruses N63, SARS-CoV-1, and -2 bind to ACE2, which is found on epithelial cells of the trachea, bronchi, and alveoli, among other tissues [133,134]. ACE2 normally has a protective role as a carboxypeptidase, modifying proteins in both the renin–angiotensin–aldosterone system, which regulates blood pressure and fluid–electrolyte balance, and the kallikrein–kinin system, which regulates vascular leakage. Disorders in both of these systems—possibly from decreased ACE2 availability due to being bound by viral particles—is thought to play a role in the pathology of severe COVID-19 [135,136]. Recombinant ACE2 as a competitive ligand has been studied as a potential therapeutic for COVID-19 [137]. Interestingly, ACE2 is also itself an ISG, and its expression is significantly induced by multiple IFN types, particularly β (I) and γ (II) [134,138]. Although one would assume this to be detrimental in the setting of SARS-CoV-2 infection (even the S protein itself has been shown to induce ACE2 expression) [139], in vitro studies have not shown that IFN induced ACE2 to enhance the virus’s replication, perhaps due to the other cellular defenses activated by the interferon response [138]. One study, however, showed that it was not the full ACE2 protein whose expression was induced by interferons, but rather a truncated isoform that did not possess the carboxylase function of ACE2 nor bind to SARS-CoV-2, and its exact function remains unknown [140,141].

### 5.2. ISG15

ISG15 belongs to the ubiquitin-like family (Ubl), which is conjugated to a variety of cellular and viral target proteins through ISGylation. ISGylation is upregulated significantly in response to IFN-I induction from viral or bacterial infection, and serves diverse biological functions, including cell cycle regulation, augmenting the immune response, and cell trafficking. ISG15 conjugation to proteins from a wide variety of viruses alters their activity and their interaction with the host machinery required for viral replication [142]. The overexpression or knockdown of ISG15 or the enzymes involved in ISGylation can directly affect viral entry, capsid assembly, budding, egress, and viral replication [143]. The PLpro from SARS-CoV-1 preferentially targets ubiquitination, but that of SARS-CoV-2 selectively targets ISG15 [127,129]. ISGylation not only protects cells against viral invasion directly by interfering with viral replication, but also indirectly through ISGylation of many host factors that respond to viral infection [144]. Although infection by influenza and Zika (ZIKV) viruses has been shown to increase ISGylation, SARS-CoV-2 PLpro mediates the de-ISGylation of ISGylated proteins, increasing the level of free (unconjugated) ISG15 in macrophages, which increases the ratio of free ISG15 to the conjugated ISG15. siRNA-mediated depletion of ISG15 or ISGylation enzymes in macrophages infected with SARS-CoV-2, ZIKV, or influenza had no effect on viral replication. However, the downregulation of ISGylation but not ISG15 was associated with the production of inflammatory cytokines, and this pattern was more significant in cells infected with SARS-CoV-2. The PLpro of SARS-CoV-2 mediates de-ISGylation, which has been shown in vitro to lead to aberrant inflammatory responses [127], which could explain the cytokine storms seen in COVID-19. One target of SARS-Co-2 PLpro is MDA5, a RIG-I-like cytosolic receptor that plays a pivotal role in the detection of cytosolic viral RNA, for which it requires ISG15 conjugation to its caspase recruitment and activation domain (CARD) [129].

### 5.3. IFIT

In humans, the IFN-induced protein with tetratricopeptide repeats (IFIT) family consists of four characterized genes: *IFIT1* (*ISG56*), *IFIT2* (*ISG54*), *IFIT3* (*ISG60*), and *IFIT5* (*ISG58*), clustered on chromosome 10. IFITs are cytosolic proteins with tandem tetratricopeptide repeats (TPRs) that form a helix–turn–helix structure that facilitates protein interactions [145,146]. The family is conserved from amphibians to mammals. The proteins lack enzymatic activity, but facilitate distinct cellular functions by binding to viral and cellular proteins [147,148,149]. TPR-containing proteins are known to influence various biological functions including cell proliferation, translational initiation, cell migration, and antiviral signaling [150]. The basal expression levels of IFITs are very low in most cell types, but they are highly induced upon activation of the IFN-I signaling pathway [151]. The IFIT family has broad antiviral activity against infection with various RNA and DNA viruses, including influenza, lymphocytic choriomeningitis virus (LCMV), herpes simplex virus (HSV), VSV, West Nile virus (WNV), cytomegalovirus (CMV) [151], adenovirus [152], and rabies virus [153]. ISG56 (IFIT1) contains nine TPRs that can form homodimeric or heterodimeric complexes with ISG56 or ISG60. In mice intranasally infected with VSV, *Ifit1*, *Ifit2*, and *Ifit3* were induced in the central nervous system. However, the infection was lethal to *Ifit2*-knockout mice [154]. ISG56 and ISG54 were induced by distinct stimuli and even differentially expressed in different cell types and tissues in mice injected with IFNβ or dsRNA at both the transcriptional and translational levels [155]. Another study showed a coordinated regulation and distinct expression pattern for ISG56, ISG49 (a mouse homologue of IFIT3) and ISG54 in the brains of mice infected with LCMV and WNV. Although, ISG56, and ISG49 were highly expressed in both distinct and common neuronal populations in the brains of LCMV-infected mice, the expression level of ISG54 was lower, delayed, and observed in fewer neuronal populations. Similar expression patterns and cellular localizations were also observed in the brains of mice infected with WNV [156]. These studies indicate the existence of a very specific expression pattern for different IFIT genes in response to particular viruses, which is also cell and tissue-dependent [155,156,157]. SARS-CoV-2 infection has been shown to induce IFITs in human lung epithelial cell lines, with IFIT1 and IFIT2 activated in A549 cells and IFIT2 and IFIT3 in Calu-3 cells.

### 5.4. IFITM

Interferon-induced transmembrane proteins (IFITMs) are a multi-member family of small proteins with strong antiviral activity. IFITMs are evolutionarily conserved across vertebrates, with five members: IFITM1, IFITM2, IFITM3, IFITM5, and IFITM10. The last two have no discernible antiviral activity; however, the others are expressed in a wide range of tissues, and play distinct roles in the innate immune response against viral infection. IFITM1 is mostly found in the cell membrane, while IFITM2 and IFITM3 are endo-lysosomal membrane-associated proteins [158]. IFITMs seem to inhibit viral infection by changing the mechanical features of cell membranes, including orchestrating a negative membrane curvature and increasing the order and rigidity of membrane lipids [159]. This allows cells to block viral infection at the entry step.

Similar to IFITs, the antiviral activity of different IFITM genes changes depending on the type of cell and invading virus. While IFITM3 inhibits influenza A virus infection more efficiently than IFITM1, the latter more effectively blocks infection by Ebola (EBOV) and Marburg (MARV) viruses. IFITM3 was shown to broadly inhibit infection by many enveloped viruses, including H1N1 influenza A, dengue virus (DENV), WNV, MARV, EBOV [160,161], yellow fever virus (YFV) [17], and MERS [162]. However, some enveloped viruses are not inhibited by IFITM proteins, including murine leukemia virus (MLV), Lassa virus (LASV), Machupo virus (MACV), and LCMV [160]. Cell entry by a pseudovirus of SARS-CoV-1 was inhibited by all three IFITMs, with IFITM3 having the strongest effect. However, when recombinant Vero E6 cells expressing ACE2 were incubated with SARS-CoV-2 and treated with trypsin, this inhibition was nullified. This suggests that IFITM proteins are unable to overcome trypsin-induced virus-host cell membrane fusion [160]. Another group cast doubt on the antiviral role of IFITMs against SARS-CoV-1, SARS-CoV-2, and MERS, and even suggested that IFITMs are hijacked to aid in efficient SARS-CoV-2 infection. In fact, IFN-I induction of IFITM proteins was shown to promote the infection of cell lines with coronavirus OC43 [163]. These conflicting findings could result from performing experiments under artificial conditions, such as using ACE-expressing cell lines, cell lines constitutively expressing IFITMs, or virus-like particles (VLPs) containing the S protein instead of wild-type viruses [164].

An association was noted between carriers of a particular IFITM3 variant and more severe symptoms of COVID-19. These patients have a single-nucleotide polymorphism in IFITM3, rs12252 [165,166], which is also associated with severe disease and death from influenza and other viruses.

### 5.5. LY6E

Lymphocyte antigen 6 complex, locus E (LY6E), also known as TSA-1 (thymic shared antigen-1), SCA-2 (stem sell antigen-2), or RIG-E (retinoic acid inducible gene E), is a member of the Ly-6/uPAR family (lymphocyte antigen-6/urokinase-type plasminogen activator receptor). It has been shown that the level of LY6E expression varies following infection by different viruses and that in some cases, it can actually promote further infection. LY6E expression was first noted to be modulated upon infection of cells with mouse adenovirus type 1 (MAV-1) and a naturally occurring oncogenic avian herpesvirus, Marek’s disease virus (MDV). Both MDV-infected chicken embryo fibroblasts [167,168] and cell lines with higher expression of Ly6E were found to be more susceptible to viral infection. Human PBMCs, CD4^+^ T cells, and THP1-cells infected with HIV had increased LY6E expression, which facilitated viral cell fusion and promoted HIV entry [169]. Transducing cells with lentivirus-expressing LY6E was also shown to increase infection by multiple enveloped viruses, including influenza A, YFV, WNV, DENV, and O’nyong’nyong virus by facilitating viral entry [169]. By contrast, LY6E-expressing cell lines were shown to resist infection by MERS-CoV, HCoV-229E, SARS-CoV, and SARS-CoV-2. In addition, *Ly6e*ΔHSC mice appeared to be more susceptible to murine coronavirus and mouse hepatitis virus A-59 [170]. LY6E also helped inhibit viral spike protein-mediated entry of HCoV-OC43 into HepG2 cells [170,171].

### 5.6. ZAP

Zinc-finger antiviral protein (ZAP), also known as PARP13, is an interferon-stimulated gene capable of recognizing CpG dinucleotides in viral RNA [172]. ZAP directly binds cellular and viral RNA through its dsRNA binding domain [173] and targets them for exosome-mediated degradation [174]. ZAPs efficiently inhibit the replication of a broad range of RNA and DNA viruses at the post transcriptional and translation stages. ZAP mediates its antiviral activity through multiple cofactors, including TRIM25 [175,176] and KHNYN, a putative ribonuclease [177]. It also interacts with various components of the 3′-5′ RNA decay complex to degrade their RNA targets. ZAPs have an N-terminal RNA binding domain with four CCCH-type zinc-finger domains crucial for antiviral activity, and a C-terminal poly (ADP-ribose) polymerase (PARP)-like domain. Both N-terminal and C-terminal domains interact with the CpG-enriched motifs of HIV [178]. Interestingly, alternative spicing results in the expression of ZAP variants with distinct C-terminal regions. These variants carry diverse and distinct innate cellular and antiviral functions against multiple viruses [179]. The long splice variant (ZAP-L) possesses a catalytically inactive PARP domain in its C-terminal, which is not included in the short isoform (ZAP-S). However, ZAP isoforms possess an identical N-terminal RNA binding domain [179]. Their unique C-terminal regions may be related to their localization in different cellular compartments, also giving them access to distinct pools of RNAs [180]. Recently, two more isoforms of ZAP have been identified, a medium-size isoform (ZAP-M) and an extra-long ZAP (ZAP-XL); these are expressed at lower levels than ZAP-S and ZAP-L. The analysis of the expression pattern of multiple members of the PARP family in HEK-293T cells stimulated with poly I:C, poly dA:dT, and 3pRNA (5′-triphosphate-modified RNA) revealed that ZAP-S, but not ZAP-L, could greatly amplify RIG-I-mediated IFNβ activation [181]. However, by using ZAP-sensitive and -insensitive HIV-I, it was shown that the C-terminal PARP domain was also important for optimal ZAP antiviral activity. The C-terminal PARP domain of ZAP-L contains a CaaX box motif that mediates the S-farnesylation post translational modification of ZAP-L. The addition of a hydrophobic group to ZAP-L facilitates its subcellular localization to the endomembrane. The CaaX-box motif facilitates ZAP-L’s interaction with TRIM-25 and KHNYN, which allows ZAP-L to recognize the CpGs of target RNAs more efficiently. This finding may partially explain why the ZAP-L isoform is a more potent antiviral compared to ZAP-S in inhibiting SARS-CoV-2 and HIV-1, whereas ZAP-S is more strongly induced in response to activation of the IFN-I signaling pathway [178]. SARS-CoV-2 hijacks the host’s translational machinery and uses PRF for the massive production of viral proteins. It has recently been shown that ZAP-S directly interacts with SARS-CoV-2 RNA and strongly impairs PRF to restrict SARS-CoV-2 infection [182]. In addition, ZAP variants have distinct basal expression levels in different cell lines. ZAP-S has a very low basal expression level, but it is highly induced in response to interferon induction, whereas ZAP-L is constitutively expressed at a high basal level, and is not significantly affected by IFN induction [175,176]. Recently it has been reported that ZAP-S also plays a role in the negative feedback that regulates IFN-I and -III expression, preventing cytokine storms that can follow viral infections. To perform this function, ZAP-S binds the 3′UTR of interferon lambda 3 (IFNL3) mRNA and destabilizes it. This effect was not seen in IFNL3 mRNA harboring a mutated AU-rich motif in the 3′UTR. ZAP-S, but not ZAP-L, was shown to interact with *IFNβ* and type-III IFNs, as well as *IFNL2* and *IFNL3* mRNA, and the expression of these cytokines was enhanced in RNA-transfected cells lacking ZAP. In response to SARS-CoV-2 infection, the levels of IFN-I, -II, and -III were significantly increased, which in turn induces ZAP expression. To counter this, the virus suppresses CpG dinucleotides to avoid ZAP-mediated viral RNA degradation by the exosome RNA-decay machinery. Optimal SARS-CoV-2 infection has been linked to higher suppression of CpG dinucleotides. Further investigations revealed that ZAP-S was significantly induced in Calu-3 cells by treatment with IFNα, IFNβ, and particularly IFNγ. The level of ZAP-L, however, showed no significant change after interferon treatment, despite ZAP-L being a stronger antiviral isoform against SARS-CoV-2 compared to ZAP-S [183].

Multiple viruses are reported to resist the antiviral properties of ZAPs. For instance, although rat fibroblasts stably expressing ZAP were significantly protected against multiple alphaviruses, including Sindbis virus (SIN), ZAP overexpression actually supported infection by HSV-1, VSV, and YFV [184]. ZAPs also do not inhibit ZIKV and DENV infection, but do protect cells against Japanese Encephalitis Virus (JEV), another flavivirus [185], indicating that the antiviral activity of ZAP is not universal.

### 5.7. OAS/RNase L System

The oligoadenylate synthase (OAS) family of proteins includes OAS1, OAS2, and OAS3, which produce 2′–5′-linked oligoadenylates upon the detection of cytosolic dsRNA. Subsequently, the 2′–5′-linked oligoadenylates act as second messengers and activate latent ribonuclease (RNAase L) to degrade cellular and viral RNA, whose remnants are then sensed by RLRs [186,187,188]. OAS polymorphisms have been seen in patients with severe COVID-19 disease [189,190,191]. An OAS-1 polymorphism was also associated with severe cases of SARS-CoV-1 [192].

### 5.8. BST2

Tetherin is a lipid raft-associated protein that is also known as CD317 and is encoded by the bone marrow stromal antigen 2 gene (*BST2*). Tetherin/BST2 is a dimer with a single alpha helix transmembrane domain and a glycosyl phosphatidyl inositol (GPI) that anchors it to lipid rafts in the cell membrane. By tethering mature virions to the surface of infected cells, BST2 prevents the spread of a wide variety of enveloped viruses, including mouse mammary tumor virus, HIV, simian immunodeficiency virus (SIV), feline immunodeficiency virus, LASV, and MARV to neighboring cells [193]. BST-2 was shown to restrict SARS-CoV-2 VLP release from infected 293T cells; however, the S protein can also co-localize with BST-2 to promote its lysosomal degradation [194,195]. Utilizing infectious cDNA clones icSARS-CoV and icSARS-ORF7abΔCoV—which lacks the ORF7ab protein—it was shown that BST-2 restriction of SARS-CoV infection is highly dependent on N-linked glycosylation of the BST2 extracellular domain. ORF7a can bind directly to the extracellular domain of BST-2 to block its glycosylation [196], and therefore BST2 could inhibit infection by icSARS-ORF7abΔCoV more efficiently than by icSARS-CoV. The transmembrane domain of BST2 is another target for viruses, as a mutant BST2 lacking the transmembrane domain could bypass the countermeasures of SARS-CoV-2 and impair its infection [197].

### 5.9. CH25H

Cholesterol-25-hydroxylase (CH25H) is a conserved intron-less gene that plays a critical role in the regulation of cholesterol through its hydroxylation to 25-hydroxy cholesterol (25HC), a natural oxysterol. The expression of genes involved in cholesterol homeostasis, including CH25H, is regulated by sterol regulatory element-binding protein 2 (SREBP-2). SREBP-2 is an endoplasmic reticulum (ER) membrane-integrated transcription factor that requires two protease cleavages in the Golgi apparatus to be in its active form. SREBP-cleavage activating protein (SCAP) and insulin-induced gene 1 (INSIG) are ER membrane-spanning proteins that monitor the level of sterols in the ER. SREBP-2 normally resides in the ER with SCAP and INSIG when the cholesterol level is normal, but when the threshold is exceeded, cholesterol binds to SCAP, which allows its binding to INSIG. The SREBP–SCAP–INSIG complex is involved in tethering SREBP in the ER and preventing its translocation to the Golgi apparatus. Similarly, 25HC binds INSIG, which, together with SCAP, holds SREBP in the ER. When the cholesterol level falls below the threshold, INSIG dissociates from SCAP and allows it to chaperone SREBP2 to the Golgi apparatus, where SREBP-2 undergoes proteolytic cleavage and is translocated to the nucleus. Subsequently, the cleaved (mature) nuclear SREBP regulates the transcription of all the genes involved in cholesterol synthesis and uptake [198,199]. 25HC blocks SREBP processing by binding to INSIG, which in turn binds SCAP. Therefore, 25HC and cholesterol act as inhibitors of SREBP processing by preventing it from transferring to the Golgi apparatus [200].

Despite being primarily recognized for its role in lipid metabolism, recently it was discovered that CH25H is also an ISG. Both CH25H and its enzymatic product, 25HC, have broad antiviral functions against a wide variety of viruses, including HSV, MHV, VSV, HIV, Ebola, Rift Valley fever virus (RVFV), YFV, and Far-Eastern tick-borne encephalitis virus [21,201]. Indeed, activation of the IFN-I pathway leads to the downregulation of genes involved in the mevalonate pathway, which ultimately blocks cholesterol biosynthesis. Interestingly, the genetic inhibition of lipid biosynthesis also establishes an antiviral state. In THP-1 cells, the downregulation of SREBP-2 resulted in the upregulation of a variety of ISGs [202].

The mechanism of 25HC antiviral activity has undergone extensive investigation. Metabolome profiling of cells that are in an antiviral state either through infection or treatment with IFN shows that 25HC inhibits multiple stages of the viral life cycle [21,203,204]. The serum concentration of 25HC is elevated in COVID-19 patients. The serum level of 25HC is also significantly increased in SARS-CoV-2 infected hACE-2 mice at five days post-infection. 25HC treatment was shown to lower the SARS-CoV-2 viral load in hACE2 mice [205].

A few groups have studied the role of CH25H in SARS-CoV-2 infection. RNA sequencing analysis of lung epithelial cells expressing ACE2 (A549-ACE2) and Calu-3 cells infected with SARS-CoV-2 showed the upregulation of IFN-I and IFN-III, which in turn resulted in the overexpression of several ISGs, including CH25H [206,207]. The inhibition of viral–host fusion is one of the mechanisms by which 25HC exerts its antiviral activity, possibly by altering the membrane shape. Lipids such as cholesterol impose a negative membrane curvature that supports viral entry. S protein-mediated receptor recognition and subsequent virus–cell membrane fusion are critical events for SARS-CoV infection. The S1 subunit initially binds to ACE2 through its RNA-binding domain, which triggers a conformational change in the S2 subunit through the formation of six-helix bundles (6HB) between the heptad repeat 1 (HR1) and heptad repeat 2 (HR2) motifs of S2. This in turn promotes viral–host membrane fusion [208]. 25HC-conjugated EK1—a peptide that inhibits 6HB formation—significantly blocks infection by SARS-CoV-2 and other coronaviruses, including HCoV-OC43 and HCoV-229E. In addition, 25HC-conjugated EK1 protects newborn mice against SARS-CoV-2 infection [209]. Utilizing a VSV-based SARS-CoV-2 pseudovirus, it was shown that the depletion of accessible cholesterol from cell membranes by 25HC prevents SARS-CoV-2 entry [207]. Interestingly, the 25HC-mediated activation of acyl-CoA cholesterol acyltransferase (ACAT) promotes the depletion of cholesterol accessible to the cell membrane. ACAT is an ER membrane-associated enzyme that plays a key role in the esterification of cholesterol, reducing its solubility in the membrane. This process facilitates the rapid internalization of accessible cholesterol from the plasma membrane and the formation of cytosolic lipid droplets. 25HC also activates the liver X receptor (LXR), which in turn activates the transcription of genes involved in cholesterol export, including ATP-binding cassette transporter A1 (ABCA1) and ATP-binding cassette subfamily G member 1 (ABCG1), or those involved in cholesterol metabolism, including cholesterol 7α-hydroxylase (CYP7A) [210]. It is known that 25HC inhibits sterol biosynthesis by binding INSIG proteins. Finally, 25HC inhibits inflammatory cytokines, including IL1β, and CH25H knockout mice were found to overexpress interleukin IL1β and be more susceptible to septic shock. This may indicate that IFN-I-induced 25HC downregulates the production of inflammatory cytokines triggered by viral infection in order to suppress the uncontrolled activation of inflammatory pathways [211].

It remains unclear through which mechanism 25HC primarily mediates its protective effect, but its broad activity makes it a promising therapeutic against multiple viruses. Rather than work against specific viral targets, 25HC and other lipid molecules alter the biophysical and biochemical features of cell membranes to inhibit the initial stages of viral infection. It would be important to determine the binding specificity of 25HC and individual 25HC analogs to INSIG1, INSIG2, LXR, and ACAT, and compare their antiviral efficacy and potential side effects.

### 5.10. FASN

In addition to genes upregulated upon IFN-I induction, there are also some downregulated by the IFN-I signaling pathway, which is the case for multiple genes involved in lipid metabolism [3,212]. This section will focus on fatty acid synthase (FASN), as it has been extensively studied by various groups. FASN is a multi-enzyme protein involved in catalyzing the synthesis of the long chain fatty acid palmitate (C16:0). Fatty acids are hydrocarbon chains of different lengths and degrees of saturation through double bonds, which are a means of storing energy [213]. They are also the building blocks of cell membranes, whose composition of fatty acids affects their biophysical and biochemical properties, including fluidity, which in turn can influence cell signaling pathways [214]. Cellular lipid homeostasis is tightly controlled by SREBP-1 and SREBP-2. The transcription of genes involved in fatty acid synthesis—including FASN, and insulin-induced glucose metabolism, are primarily activated by SREBP-1. When the concentration of intracellular lipids drops below a certain threshold, ER-resident SREBP-1 is escorted by SCAP to the Golgi apparatus, where it is activated after being subjected to two proteolytic cleavages. Subsequently, mature SREBP-1 is exported to the nucleus, where it induces the expression of genes involved in the biosynthesis of fatty acids, including FASN and acetyl-CoA-carboxylase. [198,199,215]

FASN is required for optimal growth of a wide variety of viruses [216]. The modulation of enzymes involved in fatty acid synthesis plays a pivotal role in the regulation of infection by many viruses, including human CMV [217], Kaposi sarcoma-associated herpesvirus (KSHV) [218], DENV [219,220,221], chikungunya virus [222], rotavirus [223], and hepatitis C virus (HCV) [224,225,226]. Viral infection requires optimal membrane fluidity, which leads to the clustering of receptors for cell entry [227,228,229]. The inhibition of FASN can suppress cell–cell membrane fusion and syncytia formation driven by viral proteins, impacting viral infection at multiple stages of the viral life cycle.

The efficient replication of an RNA virus is linked to its ability to modify the intracellular membrane, organize membrane phospholipids, and generate membranous organelles where viral RNA replication takes place [230]. It has been shown that different types of membranous vesicles, including ER-derived double membrane vesicles (DMVs), are the centers of viral RNA replication in the cytoplasm of infected cells, which function as viral factories. SARS-CoV-2 viral replication also takes place in distinct spherical organelles composed of vesicular membranes [231,232,233]. FASN and class III PI3 kinase (VPS34), which is involved in vesicular trafficking and autophagy, have been shown to play a critical role in the formation of DMVs. Disruption of fatty acid metabolism pathways and VPS34 activity can negatively impact SARS-CoV-2 infection. Pharmacological inhibitors of FASN and VPS34 significantly block SARS-CoV-2 replication [212,234]. The treatment of SARS-CoV-2-infected Vero-E6 cells with the VPS34 inhibitors VPS34-IN1 and PIK-III significantly blocked SARS-CoV-2 infection at different steps of the viral life cycle. Orlistat, the pharmacological inhibitor of FASN, also inhibited SARS-CoV-2 replication in Vero-E6 cells at a late stage of the viral life cycle. FASN inhibition results in the impairment of protein palmitoylation and neutral lipid synthesis. Palmitoylation is a dynamic and reversible post translational modification leading to the covalent binding of fatty acids, most commonly palmitic acid (16:0), to the cysteine residues of integral and peripheral proteins associated with the cell membrane [235,236]. Palmitoylation increases a protein’s hydrophobicity and impacts protein–protein interactions, protein trafficking, stability, and localization. Interestingly, certain viruses co-opt host palmitoyltransferases to modify viral proteins [237], since viral protein palmitoylation greatly promotes infection [235,236]. In addition, palmitoylation of the S protein is a common strategy for coronaviruses, including SARS-CoV-1 and SARS-CoV-2 [238], to improve their ability to mediate cell–cell fusion and infection. Palmitoylation also enhances the stability of the coronavirus E protein, which promotes virus assembly and infectivity. Transgenic mice expressing human ACE2 infected by SARS-CoV-2 had significantly improved survival when treated with orlistat to inhibit FASN [239]. These studies suggest that FASN inhibitors could block viral infection at multiple stages of the viral life cycle, including cell entry, replication, and budding, and may represent promising therapeutics to combat COVID-19 [233].

## 6. Perspective

The first human coronavirus discovered was HCoV-229E in the 1960s; however, coronaviruses were likely circulating in the human population for thousands of years [240], and even longer in other mammalian hosts. Unlike with some infections, humans are unable to mount a longstanding and effective memory immune response against coronaviruses and other “cold viruses”. Even our most promising SARS-CoV-2 vaccines and monoclonal antibodies have lost effectiveness over time due to the virus’s high mutation rate creating multiple new variants. However, the innate immune response is far older than the cells and antibodies that make up the adaptive immune system, and has evolved over hundreds of millions of years to defend even primitive organisms from pathogens in the environment. Our antiviral defenses, particularly the interferon response, employ myriad tactics that broadly attack viral components or focus on making host cells less hospitable. While coronaviruses may have found ways to bypass some of these strategies, their overall importance and effectiveness is highlighted by the poor clinical outcomes in SARS-CoV-2 patients with deficiencies in innate immune signaling. Our growing understanding of these mechanisms sheds light on new techniques and targets for therapeutics that could be used against SARS-CoV-2 and other emerging viruses.

## Figures and Tables

**Figure 1 pathogens-11-00538-f001:**
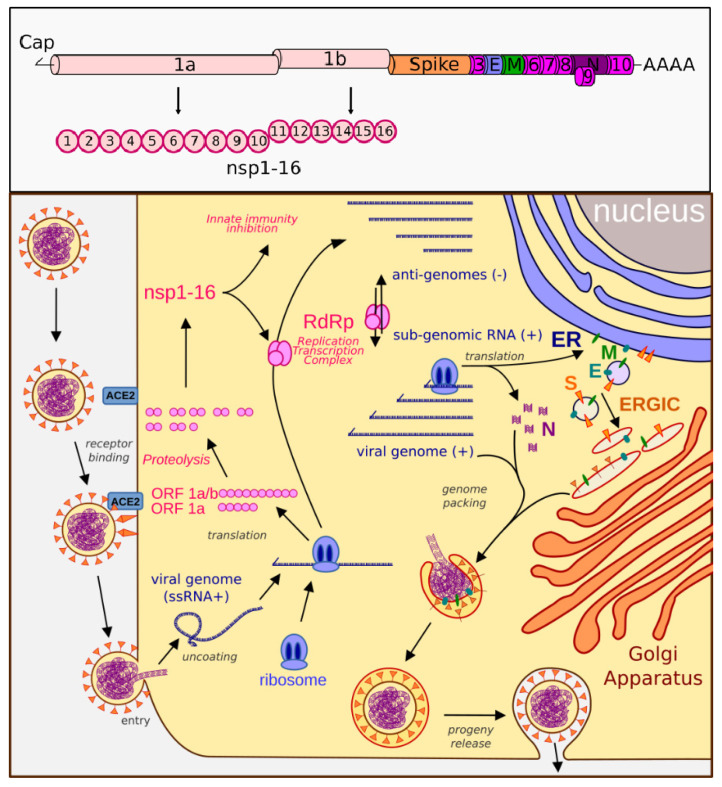
**Genome organization and life cycle of SARS-CoV-2.** Upper box: The SARS-CoV-2 genome consists of a 5′-cap, ORF1ab (encodes nsp1-16), spike protein, ORF3 (encodes ORF3a protein), E (envelope protein), M (membrane glycoprotein), ORF6 (ORF6 protein), ORF7 (ORF7a and ORF7b proteins), ORF8 (ORF8 protein), N (nucleocapsid), ORF10 (ORF10 protein), and a 3′-polyA tail. Lower box: SARS-CoV-2 binds the ACE2 receptor and fuses with the cell membrane. The plus-sense viral RNA genome is released and directly translated into polyproteins 1a and 1ab. These are cleaved to form the nsp proteins and RNA-dependent RNA polymerase (RdRp). RdRp produces the genome, anti-genome, and sub-genome copies of mRNA that encode the remaining viral proteins, including the nucleocapsid (N), membrane (M), envelope (E), and spike (S) proteins. N assembles with the genome copies while the other proteins are modified in the ER–Golgi-intermediate compartment (ERGIC), which then come together to form a mature virion that is released by exocytosis.

**Figure 2 pathogens-11-00538-f002:**
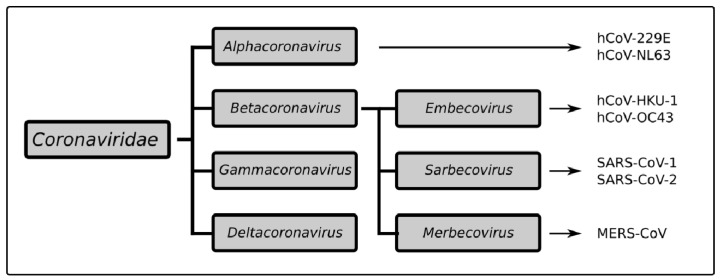
**Taxonomy of the *Coronaviridae* family.***Coronaviridae* is divided into four groups of viruses: *alphacoronaviruses* include hCoV-229E and NL-63. The *betacoronavirus* group includes three subgroups: the *Embecoviruses* with hCoV-HKU-1 and OC43, the *Sarbecoviruses* with SARS-CoV-1 and SARS-CoV-2, and the *Merbecovirus* MERS-CoV.

**Figure 3 pathogens-11-00538-f003:**
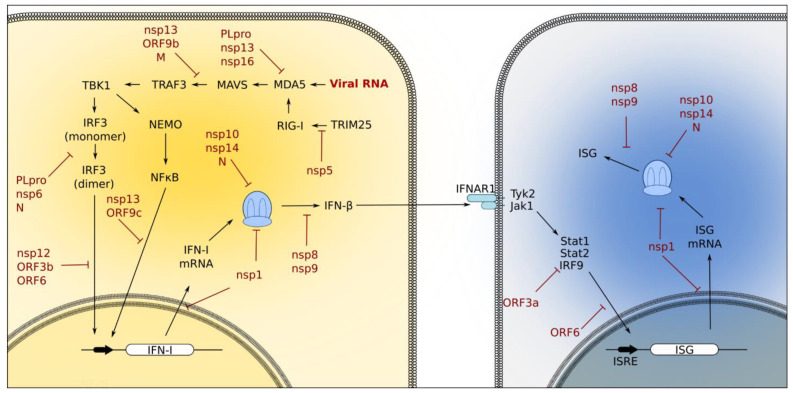
**Innate immune signaling pathways activated during coronavirus infection.** Viral RNA stimulation of the RIG-I/MDA5 pathway activates the downstream IRF3 and NF-κB pathways to induce IFN-I production. IFN-I members (such as β) then bind the IFNAR1 receptor and trigger the Jak/Stat pathway and the expression of ISGs in neighboring cells [20]. Coronavirus proteins have evolved ways to target components of these signaling pathways to prevent the interferon response. PLpro, nsp13, and nsp16 interfere with MDA5 activation. PLpro, nsp6, N, nsp12, ORF3b, and ORF6 interfere with IRF3 activation. NF-κB activation is inhibited by nsp13 and ORF9c, and Jak/Stat activation is suppressed by ORF3a. Coronavirus proteins also interfere with host cellular processes; for example, nsp1, nsp10, nsp14, and N block translation, and ORF5 blocks nuclear trafficking.

**Figure 4 pathogens-11-00538-f004:**
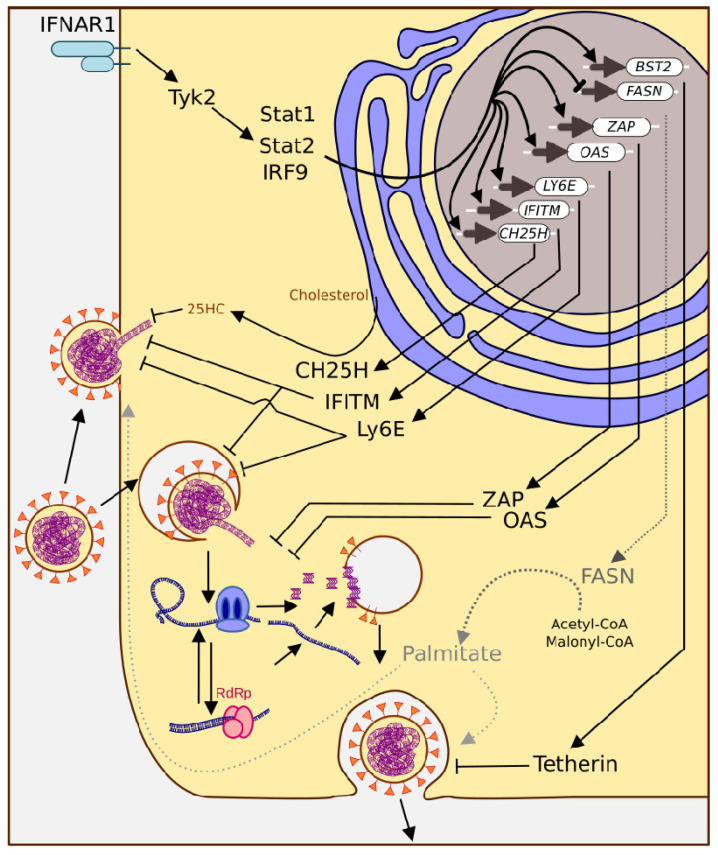
**The ISG response triggers multiple mechanisms to combat coronavirus infection.** ISGs interfere with viral entry at the plasma membrane and endocytosis (CH25H, IFITM, Ly6E), genome release (ZAP, OAS), and budding (activation of Bst2, inhibition of FASN).

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
