# Peer review of "The Evolutionary Dance between Innate Host Antiviral Pathways and SARS-CoV-2"

_pathogens, 2022, doi:10.3390/pathogens11050538_

Round 1
Reviewer 1 Report
The authors performed a well-designed review describing the current knowledge about the interactions of SARS-CoV-2 with the type I interferon (IFN-I) system. There are only few changes to be made:
- Graphical abstract: “Fasn” is difficult to read. Why did the authors show an inhibition (red) by nsp1-16 ORF3-9 from the right side of the figure (representing one cell) to the left side of the figure (representing another cell)?
- Please add a figure illustrating the genome structure of SARS-CoV-2 (nsp1-16, S, M, N, ORFs, …)
- Lines 90, 386: The authors state that there are “300 ISGs” of “few hundred ISGs” and cite references from 2011 and 2012. However, cell culture experiments demonstrated that more than 1000 genes are induced by IFN-α stimulation and that even 10% of the human genome might be regulated by IFN-I (please see Schoggins et al., 2019: Interferon-Stimulated Genes: What Do They All Do?). The authors should include the current knowledge about genes induced by IFN-I.
- Lines 88, 207: IFN-I not only acts in a paracrine but also in an autocrine fashion.
- Lines 309-311: Please specify direct and indirect actions of the M protein (SARS-CoV-1 and SARS-CoV-2). Please add current literature such as: Sui et al. Frontiers Immunology, 18 May 2021. SARS-CoV-2 Membrane Protein Inhibits Type I Interferon Production Through Ubiquitin-Mediated Degradation of TBK1. Zheng et al. Signal Transduction and Targeted Therapy (2020) 5:299. Severe acute respiratory syndrome coronavirus 2 (SARSCoV-2) membrane (M) protein inhibits type I and III interferon production by targeting RIG-I/MDA-5 signaling. Siu et al. J. Biol. Chem. 2009;284:16202–16209. Severe acute respiratory syndrome coronavirus M protein inhibits type I interferon production by impeding the formation of TRAF3.TANK.TBK1/IKKepsilon complex. Zhang et Front Cell Infect Microbiol. 2021;11:766922. Severe Acute Respiratory Syndrome Coronavirus 2 (SARS-CoV-2) Membrane (M) and Spike (S) Proteins Antagonize Host Type I Interferon Response.
- Line 312: Please introduce the abbreviations “G3BP1” and “G3BP2”.
- Lines 318-321: 76% + 34% = 110%! Please correct.
- Lines 380-381: Please specify interaction of ORF9c with LRX1, F2RL1 and NDFIP2 (introduce abbreviations)
- Lines 427-438: SARS-CoV-2 PLpro mediates “de-ISGylation” not ”deubiquitination”. Please correct “SARS-CoV2” (line 427), “levers” (line 431), “de-ISGlytion” (line 431), “de-ISGlation” (line 433), “SARS-Co-2” (line 435). The authors state that “lower levels of ISGylation but not ISG15 was associated with decreased production of inflammatory cytokines” and that “PLpro of SARS-CoV-2 mediates de-ISGylation, ... , which could explain the cytokine storms seen in COVID-19.” This contradiction should be explained à role of proinflammatory macrophages…(see reference 122)
- 3 IFIT: The authors describe IFIT proteins: IFIT1 (ISG56), IFIT2 (ISG54), IFIT3 (ISG60) and IFIT5 (ISG58). These are human proteins. In line 459 they describe a study investigating also ISG49, which is a mouse orthologue for IFIT3 (ISG60). This should be explained. The respective study (reference 150) also investigated ISG56. The terms “distinct expression pattern” (line 459) and “expression of these IFIT genes was also different” (line 460) should be specified.
- The authors have to check the consistency of their abbreviations. Please correct the following words: line 78 “type II IFN” (“type-II IFN”); line 97 “IFN-1”; line 200 “IFN III”; Figure 3 and lines 203-212 “IRF3” vs. “IRF-3”, “NF-κB” vs “NFκB”; line 215 “PAMPS”; line 228 “kockout”; line 257 “ubquitination”; line 309 “-CoV2”; line 376 “IFN-1”; Table 1 “SARS-Cov-2” (multiple times), “IFIT-I”, “Viiperin” (2 times) and “NfκB” ( 2 times); References in the second line “(86, 99-101) (103)”; line 453 “TPRS; line 459 “ISG-49” and “ISG-54” (delete “-“); line 542 “5’-triphophate-mpfified RNA”; line 555 “-1PRF” (delete “-1”); line 583 “5.7.(. OAS)/RNase L system”; line 588 “RLRs(180-182)”; lines 721-722 “SARS-COV-2”; line 732 “infection. (229, 230)”; line 734 “SARS-Cov-1”
- Line 446: Please replace “perform” by “influence”.
- Please check all references: In Table 1 reference 103 is given for nsp14 but the respective study investigated nsp1 and did not investigate interactions of nsp14 with IRF3 or IFIT1.
Reviewer 2 Report
In this manuscript, the authors comprehensively reviewed the publications on interactions of SARS-CoV-2 with the host innate immune pathways since the pandemic of COVID-19 started. It started with the overview of the phylogenetic classification of SARS-CoV-2 within the Coronaviridae family and extended it to a variety of aspects of innate immune systems, including germline-encoded pattern recognition receptors and type I IFN-oriented innate signaling. The manuscript provided significant information and was also well-organized. Another strength of this manuscript was that it pinpointed the interactions of viral composite with specific element host innate immune systems, for example, NSP1 interfered with the IFN-I pathway and inhibits host protein expression et c.. However, the manuscript will benefit from address several comments below:
- Many abbreviations were generated and included. It was suggested to have a list of all abbreviations in the manuscript.
- Although the manuscript focused on type I IFN, the manuscript also included other components of the innate immune signaling. However, it was not clear how the interactions of IFN-I with SARS-CoV-2 were associated with the severity of the disease and/or long COVID, or even with the asymptomatic infections. Since this virus caused so many human infections, this aspect of information will be significant.
- Many papers included and described in this manuscript were about in vitro studies. However, the contributions of different innate immune signaling to the virus infection would be less significant if the studies were limited in the in vitro studies. Therefore, the manuscript will provide more valuable information of more in vivo studies were included and reviewed.
- Some of the words may need more edits, for example, “bind to” and “bind” have been used several times and seems exchangeable; “hospitable” ….
